# A Comparison of Magnetic Resonance Imaging Methods to Assess Multiple Sclerosis Lesions: Implications for Patient Characterization and Clinical Trial Design

**DOI:** 10.3390/diagnostics12010077

**Published:** 2021-12-30

**Authors:** Ewart Mark Haacke, Evanthia Bernitsas, Karthik Subramanian, David Utriainen, Vinay Kumar Palutla, Kiran Yerramsetty, Prashanth Kumar, Sean K. Sethi, Yongsheng Chen, Zahid Latif, Pavan Jella, Sara Gharabaghi, Ying Wang, Xiaomeng Zhang, Robert A. Comley, John Beaver, Yanping Luo

**Affiliations:** 1The MRI Institute for Biomedical Research, Bingham Farms, MI 48025, USA; davidutriainen@gmail.com (D.U.); sethisea@gmail.com (S.K.S.); 2Department of Radiology, Wayne State University, Detroit, MI 48201, USA; fb6206@wayne.edu (K.S.); zahidlatif@wayne.edu (Z.L.); pavan.jella@wayne.edu (P.J.); ying.wang11@wayne.edu (Y.W.); 3Department of Neurology, Wayne State University, Detroit, MI 48201, USA; ebernits@med.wayne.edu (E.B.); ys.chen@wayne.edu (Y.C.); 4SpinTech Inc., Bingham Farms, MI 48025, USA; 5MR Innovations Inc., Bingham Farms, MI 48025, USA; sg.ele.eng@gmail.com; 6MR Medical Imaging Innovations India Pvt. Ltd., Hyderabad 500081, India; pvin123@gmail.com (V.K.P.); kiran.yerramshetty@gmail.com (K.Y.); nukalaprashant@gmail.com (P.K.); 7AbbVie Inc., North Chicago, IL 60064, USA; xiaomeng.zhang@abbvie.com (X.Z.); robert.comley@abbvie.com (R.A.C.); john.beaver@abbvie.com (J.B.); yanping.luo@abbvie.com (Y.L.)

**Keywords:** multiple sclerosis, chronic white matter lesions, demyelinating/inflammatory lesions, quantitative magnetic resonance imaging

## Abstract

Magnetic resonance imaging (MRI) is a sensitive imaging modality for identifying inflammatory and/or demyelinating lesions, which is critical for a clinical diagnosis of MS and evaluating drug responses. There are many unique means of probing brain tissue status, including conventional T1 and T2 weighted imaging (T1WI, T2WI), T2 fluid attenuated inversion recovery (FLAIR), magnetization transfer, myelin water fraction, diffusion tensor imaging (DTI), phase-sensitive inversion recovery and susceptibility weighted imaging (SWI), but no study has combined all of these modalities into a single well-controlled investigation. The goals of this study were to: compare different MRI measures for lesion visualization and quantification; evaluate the repeatability of various imaging methods in healthy controls; compare quantitative susceptibility mapping (QSM) with myelin water fraction; measure short-term longitudinal changes in the white matter of MS patients and map out the tissue properties of the white matter hyperintensities using STAGE (strategically acquired gradient echo imaging). Additionally, the outcomes of this study were anticipated to aid in the choice of an efficient imaging protocol reducing redundancy of information and alleviating patient burden. Of all the sequences used, T2 FLAIR and T2WI showed the most lesions. To differentiate the putative demyelinating lesions from inflammatory lesions, the fusion of SWI and T2 FLAIR was used. Our study suggests that a practical and efficient imaging protocol combining T2 FLAIR, T1WI and STAGE (with SWI and QSM) can be used to rapidly image MS patients to both find lesions and study the demyelinating and inflammatory characteristics of the lesions.

## 1. Introduction

Multiple sclerosis (MS) is a neurodegenerative disease characterized by a wide range of symptoms, continued progression over time, autoimmune responses and vascular dysfunction. Magnetic resonance imaging (MRI) is a sensitive imaging modality for identifying inflammatory and/or demyelinating lesions, which is critical for the clinical diagnosis of MS and evaluating drug responses. The ability to detect the hallmark periventricular Dawson’s fingers using T2 weighted imaging (T2WI) allows the radiological diagnosis of MS [1]. The administration of T1-shortening contrast agents allows the detection of blood–brain barrier breakdown to reveal acute lesions. However, there are many other unique means to probe brain tissue status, including myelin water fraction imaging (MWF) [2,3], diffusion tensor imaging (DTI) [4], magnetization transfer ratio (MTR) [5,6], proton spin density (PSD) mapping, T1 mapping, T2* mapping and quantitative susceptibility mapping (QSM), to name a few [7,8,9,10,11,12,13,14,15]. All these methods attempt to increase the specificity of the disease diagnosis and to reflect key underlying mechanisms to study the etiology of MS. Previous multi-parametric studies have attempted to find correlations between imaging measures for a given lesion or set of lesions [16,17,18], but none that we are aware of have combined most of the MR imaging sequences into a single investigation.

In this work, we also introduce the use of Strategically Acquired Gradient Echo (STAGE) imaging to study lesion properties and compare different MRI measures for lesion visualization and quantification relative to tissue water content and magnetic susceptibility. STAGE provides quantitative maps for both proton spin density, i.e., water content, and for susceptibility (via T2* maps and quantitative susceptibility mapping (QSM)). These two measures are studied in this work as well as the above-mentioned quantitative results for T1, DTI, MTC, and MWF. The use of QSM and FLAIR opens the door to studying the demyelinating and inflammatory characteristics of the lesions. In total, 20 patients and 10 healthy controls were imaged twice to evaluate the repeatability of these measures and to correlate the various qualitative and quantitative clinical and functional measures. In particular, we used the Expanded Disability Status Scale (EDSS), Functional System Score (FSS) and Multiple Sclerosis Functional Composite (MSFC) scores to evaluate the patients clinically. Our final goal is to recommend a practical and efficient rapid imaging protocol to reduce patient burden.

## 2. Materials and Methods

### 2.1. Patient Demographics

The data were acquired under local IRB approval at Wayne State University (Detroit, MI, USA) and all the patients signed an informed consent form. Twenty (20) relapsing-remitting MS patients and ten (10) age-matched HCs were included in the study. All the MS patients were imaged twice six months apart and HCs were imaged at baseline and two weeks later. The patients were aged 37.4 ± 9 years with 15 females and 5 males, while the HCs were aged 35.4 ± 12 years with 6 females and 4 males. The disease duration in MS patients ranged from less than a year to 22 years, with a mean disease duration for the whole MS cohort of 6.1 ± 5.0 years. The Functional System Score (FSS) ranged from 0–3 and the Disability Status Scale (EDSS) scores ranged from 0–4, with a median EDSS score of 1.5 at both time points. The patients completed EDSS and the Multiple Sclerosis Functional Composite (MSFC) testing prior to both imaging time points. If patients were receiving medical treatment, any disease modifying drugs were recorded and are presented herein along with other patient demographics in Appendix A.

The inclusion criteria for patients were as follows: at least 18 years of age, able to understand and sign the consent form, a confirmed diagnosis of MS according to the revised McDonald criteria [1], a Kurtzke EDSS of less than 6.0, in good health with the exception of MS, and neurologically stable with no evident relapse or corticosteroid treatment in the 30 days prior to the screening visit. The exclusion criteria for patients were as follows: pregnant or nursing, MRI-contraindicated, claustrophobic or unable to lie still for at least one hour, a history of major illness such as chronic renal disease, a prior known neurological disorder (other than MS), history of substance abuse, progressive MS diagnosis, incidents of seizure or unexplained blackouts within six months of screening, known sensitivity or allergy to the Gadolinium (Gd)-based contrast agent Gadavist, creatine level > 1.4 mg/dL, any significant brain abnormality other than MS or initiation or switching of medication within the 6 months following screening.

The inclusion criteria for the HCs were as follows: aged between 18 and 65 years, neurologically stable, in good health based upon the results of their medical history, physical examination and vital signs. The exclusion criteria for HCs were as follows: diagnosis of MS in the past, findings on the brain MRI scan indicating any clinically significant brain abnormality, a positive screen for non-prescribed drugs or alcohol, history of drugs and alcohol abuse within 6 months prior to screening or history of seizure disorder or unexplained blackouts within six months prior to screening.

### 2.2. MR Imaging Protocol

All the patients were imaged on a 3T Siemens VERIO (Erlangen, Germany) at Wayne State University with a 12 channel head-and-neck coil. The MRI protocol included: magnetization prepared rapid gradient echo (MP-RAGE), pre- and post-contrast T1-weighted imaging (T1WI) and T2-weighted imaging (T2WI), 3D T2 fluid attenuated inversion recovery (FLAIR), DTI, MTC, MWF, susceptibility weighted imaging (SWI) and STAGE imaging (SpinTech Inc., Bingham Farms, MI, USA) [19,20,21,22]. The imaging parameters for each sequence are given in Appendix A. All the sequences were acquired prior to Gd injection except the post-contrast T1WI and DTI. All the fields of view were set to the same value and all the data were collected in a transverse mode except for the T2 FLAIR sequence, which was collected sagittally. All the resolution dimensions were multiples of 0.67 mm. The T2 FLAIR sequence was reformatted into transverse slices with a 0.67 mm effective slice thickness and 0.67 × 0.67 interpolated in-plane resolution. Three adjacent slices were added to create a 2 mm-thick slice to match the transverse-collected STAGE data. All the data were co-registered using SPIN software (SpinTech Inc., Bingham Farms, MI, USA).

### 2.3. Image Processing

The STAGE images were processed using custom MATLAB-based software in order to generate: SWI, T2* maps, QSM data [23], T1 maps, PSD maps, and SWI-FLAIR. The proposed QSM reconstruction algorithm, referred to as scSWIM (structurally constrained susceptibility weight imaging and mapping), performs an L1 and L2 regularization-based reconstruction in a single step [24]. It uses the unique and enhanced contrast of STAGE imaging to extract reliable geometry constraints. These constraints include segmented deep gray matter structures, vessels and other high-susceptibility regions. Furthermore, the edges of white matter and gray matter are used as prior information. Iterative susceptibility weighted imaging and mapping (iSWIM) is used as an initial starting point to the scSWIM process since it is faster and provides an initial susceptibility map with well-defined veins [24].

The software, DTI studio (John Hopkins University), was used to generate fractional anisotropy (FA), radial diffusivity (RD), and apparent diffusion coefficient (ADC) maps [25]. The magnetization transfer ratio (MTR) was calculated from the MTC acquisition as the ratio (MTCoff-MTCon)/MTCoff. MWF was calculated using an in-house MATLAB (Math-Works, Natick, MA, USA) script [26]. Multi-echo T2 relaxation curves were plotted for each voxel. The distributions of the T2 relaxation components were generated using 200 logarithmically spaced T2 values ranging from 10 to 2000 ms. MWF was calculated as the sum of the T2 relaxation times from 10 to 40 ms normalized to the sum of the T2 times from 10 to 2000 ms. Higher MWF values represent higher myelin content. Due to SNR and time constraints, we ran the MWF sequence with a thicker slice than the other modalities. Since the MTR and MWF images could not be registered to the T2WI/FLAIR images exactly, the lesions were viewed side-by-side with T2W imaging and T2 FLAIR and then drawn manually on the MTR and MWF images for all the slices in which they appeared.

Three-dimensional T2 FLAIR images were evaluated for white matter hyper-intensities (WMH), which were quantified into lesion load using a semi-automated tool (SPIN software, SpinTech Inc., Bingham Farms, MI, USA) and later confirmed by a radiologist. All the lesions were then evaluated for their appearance in QSM and SWI high-pass filtered phase data in the baseline images. Those that could be visually observed (positive susceptibility) compared to the surrounding white matter were manually drawn as a lesion region of interest (ROI) and classified as QSM positive (QSM+) lesions. These lesions were then manually drawn according to the contrast in each modality (the ROIs were not copied between each modality as the appearance of the lesion varied between each modality). Normal-appearing white matter (NAWM) regions of the same size were then drawn on the contralateral side of the brain across different slices as close to being symmetric with the lesions as possible. NAWM ROIs were also drawn in the healthy control sampling four different regions in both hemispheres within the frontal, temporal, and parietal lobes (Figure 1).

The T2W and T2 FLAIR lesions that were observed but lacked contrast in the QSM and SWI phase were then drawn separately and classified as QSM-negative (QSM−) lesions. The lesion boundaries from the T2W images were copied onto the susceptibility maps and the QSM-negative lesions were manually drawn and measured across all other modalities. Post-contrast T1W lesions that showed enhancement at either time-point were labeled and ROIs were placed across all modalities to measure changes in lesion signal and size. If any T2 FLAIR WMHs were observed in the control population, these were drawn and measured across all modalities similarly. All the MS lesions that appeared in the baseline scans were redrawn at the same position in the follow-up scans to account for potential changes in lesion boundaries. New lesions and lesions that disappeared at the second time point were noted when observed.

The HC data were evaluated for test–re-test reliability. The ROIs of at least 100 pixels were drawn in the NAWM, thalamus, and cerebrospinal fluid (CSF). For each ROI, the mean and standard deviation of the intensity in each image type were measured. SPSS version 22 (IBM, Armonk, NY, USA) was used to perform a t-test for consistency between baseline and follow-up for each measure.

### 2.4. SWI-FLAIR

A fusion image of T2 FLAIR with SWI was created. This was performed using the high-pass filtered phase data and creating a mask that highlighted high-susceptibility regions. To generate the SWI-FLAIR data, the phase mask was multiplied into the co-registered T2 FLAIR image eight times (similar to what is done in creating SWI [27]). The phase mask represents changes in susceptibility from iron content or demyelinated white matter. The advantage of combining these two images is the ability to easily visualize lesions with iron or demyelinated white matter. The presence of any susceptibility change can be compared with MWF to ascertain whether the loss of myelin correlates with increases in susceptibility. If there were changes in both MWF and SWI (i.e., SWI-FLAIR lesions show lowered signal from the susceptibility changes) then these lesions could be considered both inflammatory and demyelinating. However, if there were no changes in SWI-FLAIR relative to T2 FLAIR, the lesions could be considered inflammatory but not demyelinating.

### 2.5. Statistics

When comparing the results over time, the Z-score was calculated by finding the change in mean intensity within the lesion across the two time points and dividing by the standard error of the mean. All the lesions in patients with MS were grouped by QSM+ and QSM−, and a paired sample *t*-test was used to analyze the differences in lesion volumes between two time points for the two groups of lesions. Pearson correlation analysis was performed to assess the test-retest repeatability of the methods on all quantitative indices measured on the healthy controls. Various imaging measures were correlated with each other using least-squares regression and the *p*-values were calculated from the r-value. A *p*-value less than 0.05 was considered significant. The scores from the clinical tests were correlated using SPSS (International Business Machines Corporation, SPSS statistics for Windows, version 24.0, Armonk, NY, USA) against subjects with high-susceptibility lesions (>30 ppb). Spearman’s correlation and a t-tailed test were used with a 95% CI.

### 2.6. Clinical and Functional Measures

The EDSS score ranges from 0 to 10, which helped to measure and monitor the disability level over time of the MS subjects [28]. FSS ranges from 0 to 6, measuring disability based on major central nervous system outcomes. FSS involves the use of pyramidal, cerebellar, brainstem, sensory, bowel/bladder function, visual and cerebral functions. MSFC is another multidimentional scoring system used to evaluate disability levels in MS subjects [29]. MSFC is comprised of three clinical components: nine-hole peg test to assess arm and hand functionality, paced auditory serial addition test (PASAT-3) to assess cognitive function, and a timed 25 foot walk test for leg function.

## 3. Results

### 3.1. Lesions in MS Subjects

Eighteen (out of twenty) MS subjects had lesions at both time points. An example image of a typical MS lesion appearance across all modalities and their image quality is shown in Figure 2. Two of the MS subjects had no visible lesions in T2 FLAIR. The remaining eighteen MS subjects had visible deep and periventricular WMHs in T2 FLAIR. The lesion count in MS subjects varied from 2 to 84 lesions in the 18 subjects. Six subjects had lesion loads between 0.1 and 1.0 cc, eight subjects had lesion loads between 1.0 and 10 cc, and four MS subjects had diffuse WMH volume that exceeded 25 cc as measured from the T2 FLAIR data. The overall lesion burden did not change between time points in the six-month period between scans for any of the subjects.

### 3.2. Lesions in HC Subjects

Eight of the ten HC subjects had no visible white matter lesions in the T2 FLAIR data; however, two had subcortical WMHs. One control (aged 58 years) had a lesion volume of 0.5 cc (13 tiny lesions), as seen in the T2 FLAIR image, and it remained unchanged at follow-up. This HC subject was diagnosed with retinopathy and patients with retinopathy are more likely to have white matter lesions [30]. Another HC (aged 56 years) had a lesion volume of 0.1 cc (5 tiny lesions), which was unchanged at the second time point. Studies suggest that there is an increase in the number and volume of WMH with age in HC [31,32]. The other eight HC subjects were younger than this individual (<43 years). As expected, imaging measures derived from different MRI sequences did not show significant differences (adjusted *p* > 0.80) in the HC test–retest over the 10 days between scans. There was no significant change in the different sequences between baseline and follow up scans (*p* > 0.14 in all cases).

All the lesions that were observed in QSM also appeared as WMHs in T2 FLAIR images but not all WMH lesions in T2 FLAIR were observed in QSM. One subject showed changing lesions over time (one subject developed two new lesions at the second time point, one of which had slight contrast enhancement at the second time point). Appendix A shows the average lesion measurements in MS subjects; this data reveals that there was no significant difference in the average NAWM tissue measurements between MS patients and HC subjects.

### 3.3. QSM ± Lesions

Lesions were considered either QSM-positive or QSM-negative. The QSM-positive lesions showed a mean susceptibility varying from 0 to 85 ppb (parts per billion). The QSM-negative lesions showed a slightly negative or near-zero susceptibility, denoting similar susceptibility values seen in NAWM. A total of 384 QSM-positive lesions were found in 18 subjects and 94 QSM-negative lesions were found in 11 subjects 7 subjects had only QSM-positive lesions). For the MTR data, an ROI was drawn just outside the lesion to determine whether there was any change in the tissue properties surrounding the lesions between the time points. However, no significant changes (*p*-value = 0.35) were observed in the peripheral tissue values longitudinally. No differences were seen between the two time points for the QSM and MTR results for either the lesions or the contralateral NAWM (*p*-value = 0.33). Two MS subjects had large, diffuse lesions and the drop in volume in those two cases may be due to partial volume effects and systematic drawing error as contrast in diffuse lesions can often be low. The measurements of the smaller lesion appeared consistent from scan 1 to scan 2.

### 3.4. Gd-Enhancing Lesions

Gd-enhancing lesions were only seen in one of the MS subjects. This subject had changes in MTR between time points in four lesions which showed T1W Gd enhancement. The average drop between time points was 0.1 for MTR (a 20–25% reduction). MTR did not show predictive power for two lesions that would become acute in the time point 2 scan. Two other lesions that showed enhancement at time point 1 but did not enhance at time point 2 showed nearly normal contrast relative to surrounding tissue in time point 1 in MTR, but a drop in MTR was observed by time point 2.

### 3.5. Correlation between Different Imaging Sequences

Since there was no significant change in the different measures between the two time points, the lesion and NAWM values were averaged for all the figures. The T2 FLAIR and T2W images showed similar changes across all lesions, as shown in Figure 3. However, lesion volume in T2 FLAIR appeared to be larger than that in T2W images as T2 FLAIR includes diffuse lesions which are more difficult to see in T2W images. Figure 3 also shows the correlation in intensity for lesions across MTR, ADC, T2W, and T2 FLAIR. All four of the QSM, MTR, T2W, and FA maps showed significant correlations with each other. The STAGE data were used to calculate the T1 and water content values (referred to as T1 map and PSD map). Both the T2W signal (Figure 3e) and the T1 values (Figure 3f) correlated with water content.

The relative QSM intensity (the actual QSM value minus the surrounding NAWM value) for lesions and NAWM is shown in Figure 4 (relative susceptibility was used since there were lesions with values close to 0 ppb although the surrounding NAWM values were closer to −50 ppb). As expected, if there was an increase in water content in the tissue, the lesions showed a longer T1 than normal tissue, increasing from roughly 900 ms (NAWM) to 1400 ms (see Appendix A). The water content (the PSD values) also increased from roughly 2100 (NAWM) to 2500 in arbitrary units.

The lesions and the NAWM regions showed a clear separation in signal and/or tissue properties for all sequences. For MS lesions that appeared clearly in QSM relative to NAWM, their intensities showed a strong separation between MTR, T2WI, FA and ADC (see Figure 4). The DTI measures, including FA, ADC and RD are plotted against each other in Figure 5 and showed a strong correlation with one another. The SWI-FLAIR images were used to show which lesions had a significant positive susceptibility change (increased paramagnetism) of the WM, potentially signifying demyelination (Figure 6). Typically, the T2 FLAIR lesions extended beyond the boundary of the QSM or SWI-phase visible lesions.

MWF had a thicker slice than other modalities and lower resolution, making it difficult to see smaller lesions; however, larger lesions could still be easily visualized. In total, 162 of 384 (42.2%) QSM-positive lesions and 29 of 94 (30.8%) QSM-negative lesions were visible in the MWF images. An example image showing MWF compared to other modalities is shown in Figure 7. The MWF correlated with susceptibility for QSM visible lesions, indicating that both methods were sensitive to loss of myelin fibers in white matter lesions (see Figure 8, *p*-value = 0.01). It can be seen from Figure 2 and Figure 7 that lesion appearance differs between modalities. FLAIR lesions are diffuse, lesions in modalities such as T1, T2WI, MTR are smaller and QSM lesions are local and have different volumes.

### 3.6. Correlation between Different Clinical and Functional Measures

We plotted the number of QSM lesions exceeding a threshold of 20 ppb and 30 ppb (potentially representing more demyelinating lesions) versus EDSS; the number of QSM lesions dropped from 383 to 142 and 76, respectively. There was a trend for increased lesion count with increasing EDSS scores, although for a subset of these lesions there was still no correlation with EDSS (Figure 9). Our data show two different populations, one group where EDSS increases with the number of lesions (lesion load) and another where the number of lesions does not correlate with EDSS. Four out of the twenty MS subjects had QSM positive lesions with an average mean susceptibility higher than 50 ppb. Three of those subjects had an EDSS increase of 1.5 over the 6 month time interval. The other subject had an EDSS increase of 0.5 (see Appendix A). No strong correlation was found between the disability outcome and the QSM lesion count. The mean FSS and MSFC scores were not significantly different between the baseline and the follow-up scan. Furthermore, the FSS and MSFC scores did not show any significant correlation with lesion count and lesion volume. No correlation was observed between the clinical scores themselves either (EDSS vs. FSS vs. MSFC).

## 4. Discussion

We first investigated the correlation of the various potential MRI measures to ascertain which, if any, were dependent on each other. The purpose behind this was to determine whether some scans are redundant, not providing truly novel information that cannot be ascertained from the other imaging modalities. In this regard, we found that MTR, T2WI, and ADC all correlated strongly with the T2 FLAIR signal intensity. Of all the diffusion measures, ADC had the strongest correlation with T2WI data. Further, FA, RD, and ADC all correlated with each other. The correlation of these parameters (except for susceptibility) with T2WI suggests that they are all sensitive to the amount of water present in the tissue. This is reinforced from the plot of T2W imaging data versus the proton density. The linear correlation of many of these variables from what are likely chronic lesions suggests that these measures are not independent at this stage of their development.

Both T2 FLAIR and T2W imaging showed the most lesions and the clearest lesions out of all methods, while DTI measures showed lesions the worst because of their low resolution and poorer contrast-to-noise ratio. The reason all the DTI measures correlated with each other is likely to have been the presence of chronic lesions with high water content. Although DTI can be sensitive to some pathological mechanisms, the presence or absence of lesions is not dependent on DTI data. Similarly, the increases in water content (as shown in the proton spin density weighted imaging data) in chronic lesions leads to an increase in the signal from the T2 component. Generally, water content should be considered when measuring MWF changes in acute lesions [33], and it is also believed that the change in total water content could cause the decrease in MWF that was observed after iron extraction [3]. Further, it has been shown that a change in MWF does not necessarily reflect a change in myelin content and that the sensitivity of myelin water fraction to changes in iron content in the brain has far reaching consequences [34].

For this cohort of MS subjects, it is likely that most lesions are chronic (there were few enhancing lesions) and chronic lesions tend to have the highest water content and tissue damage may be the most severe. One might even expect that a higher level of atrophy will exhibit higher signal in FLAIR and T2 because of the increased water content. Therefore, it is not surprising that many of the imaging measures such as DTI, T2WI, MWF, and MTR all correlate with the proton spin density in these lesions. The latter is a direct measure of the total water content in the tissue as extracted from the STAGE data. Under these circumstances, these other measures may offer no new information for diagnosing or monitoring changes in lesion size or visibility.

The relative susceptibility of NAWM varied from slightly more diamagnetic in the dense corpus callosum to slightly negative or close to zero ppb in the surrounding white matter. The average susceptibility for demyelinating lesions was around 50 ppb, while the surrounding NAWM was closer to or less than zero ppb. The presence of lipid macromolecules such as myelin reduced tissue susceptibility, resulting in increased relative susceptibility in demyelinated lesions relative to NAWM, whereas the other lesions with less or no susceptibility change relative to NAWM could represent inflammatory lesions, which may have less impact on the disability scores [35,36,37]. Therefore, one might conclude that FLAIR lesions with high relative susceptibilities are likely demyelinated and inflammatory while FLAIR lesions with no relative change in susceptibility are likely inflammatory only (without the lesions having yet progressed to the demyelinated state). These two states can be differentiated using the fusion of SWI and T2 FLAIR into SWI-FLAIR. Unlike other studies [38,39] that found some lesions seen in QSM that were not seen in T2W imaging or T2 FLAIR, no such lesions were found in this study. This may be because many of these lesions are chronic, and it has been shown that chronic lesions undergo reduced susceptibility changes [40].

## 5. Limitations

There are several limitations to this study. First, all the patients received disease-modifying drugs, remained stable and lacked any relapses; therefore, it was not a surprise that within a six-month window the different imaging measures did not change. A larger time window between the scans would have been ideal. Second, the scan time was limited per sequence. Ideally, a higher resolution might have improved small lesion detection. More specifically, the MWF sequence had a high bandwidth and low resolution, making it difficult to detect small lesions. Furthermore, the SWI scan could have been run with a higher resolution on the order of 0.67 × 1 × 1.34 mm^3^ in less than 5 min per scan [21]. Third, our sample size is small, which limited the generalization of our results to a larger population. Previous studies have examined the relationship between EDSS, FSS, MSFC and MRI modalities and yielded conflicting results [41,42]. In our study, no significant correlation was found between them. A larger cohort might provide a better and strong clinical correlation. Finally, another limitation was that the location of the MS lesions and their spatial correlation with the clinical scores were not evaluated. Finally, although a conventional MS screening protocol includes spine imaging, in this paper, we did not analyze spinal MS lesions.

## 6. Conclusions

In conclusion, MRI offers a wide range of sequences that are sensitive to the structural and functional changes within MS lesions. In this study, almost all the measures were found to correlate with the high water content of the lesions. This suggests that DTI, MWF, and MTR measures do not add new information for chronic lesions that is not already seen with methods highly sensitive to water content, such as PSD and T2 FLAIR imaging. Choosing the ideal protocol would entail using sequences that provide independent pieces of information which, based on our findings, would include: T2 FLAIR for subtle lesion detection; STAGE (providing T1W and PSD weighted images along with T2*, susceptibility and PSD maps for lesion quantification; possibly with QSM as a biomarker for demyelination); and PWI for acute lesion detection. Allowing 5 min for FLAIR and PWI scans each and 10 min for with a resolution to 1 mm^3^ voxel size for STAGE, the total imaging time would be less than 20 min and still provide a comprehensive set of quantitative data for studying white matter lesions in MS.

## Figures and Tables

**Figure 1 diagnostics-12-00077-f001:**
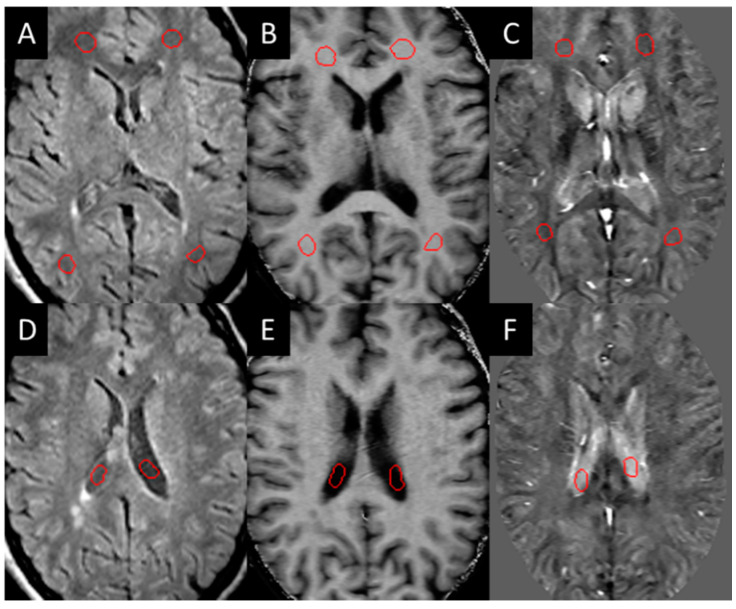
Example of NAWM ROIs (red circles in **A**–**C**) drawn on the white matter and in the CSF (red circles in **D**–**F**) for test re-test reliability: (**A**,**D**) T2 FLAIR, (**B**,**E**) MTR, (**C**,**F**) iSWIM.

**Figure 2 diagnostics-12-00077-f002:**
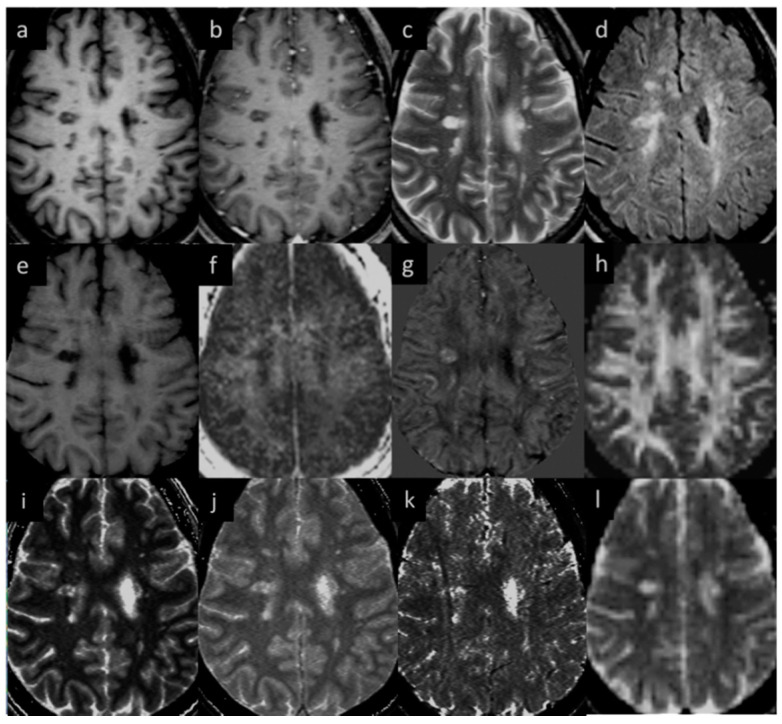
Lesion appearance in different modalities: (**a**) pre-contrast T1W, (**b**) post-contrast T1W, (**c**) T2WI, (**d**) T2 FLAIR, (**e**) MTR, (**f**) MWF, (**g**) QSM, (**h**) FA, (**i**) T1MAP, (**j**) PSDMAP, (**k**) T2STAR, and (**l**) ADC.

**Figure 3 diagnostics-12-00077-f003:**
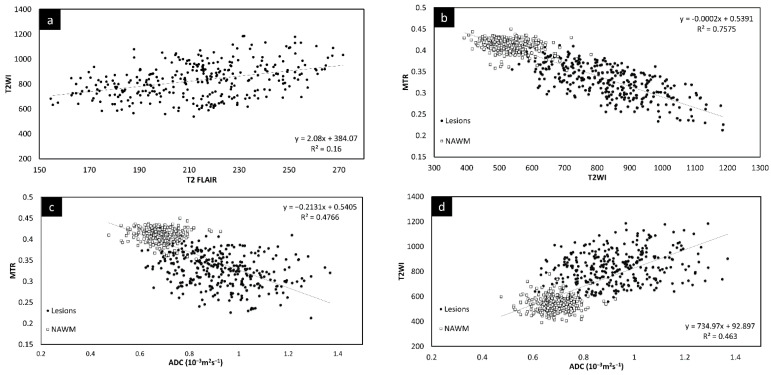
Signal intensity of T2WI versus T2 FLAIR lesions averaged across both time points (**a**). There is a linear relationship between the two modalities. Signal intensity of T2WI versus MTR lesions and NAWM averaged across both time points (**b**). The plots show that the signal intensity of the MTR data correlates with T2WI data suggesting that water content is the key driver to these changes in both measures. MTR plotted against ADC averaged across both time points (**c**). MS lesions have a significant drop in MTR and increase in ADC. T2WI signal intensity versus ADC averaged across both time points (**d**) showing lesions have a significant increase in signal for both T2WI and in ADC. T2WI signal intensity versus PSDMAP averaged across the two different time points (**e**) showing lesions have a significant increase in signal in both T2WI and in spin density. This shows they are both sensitive to the amount of water present in the tissue. Signal intensity of the R1 map (**f**) versus PSDMAP averaged across the two different time points shows a clear separation of healthy controls and MS subjects.

**Figure 4 diagnostics-12-00077-f004:**
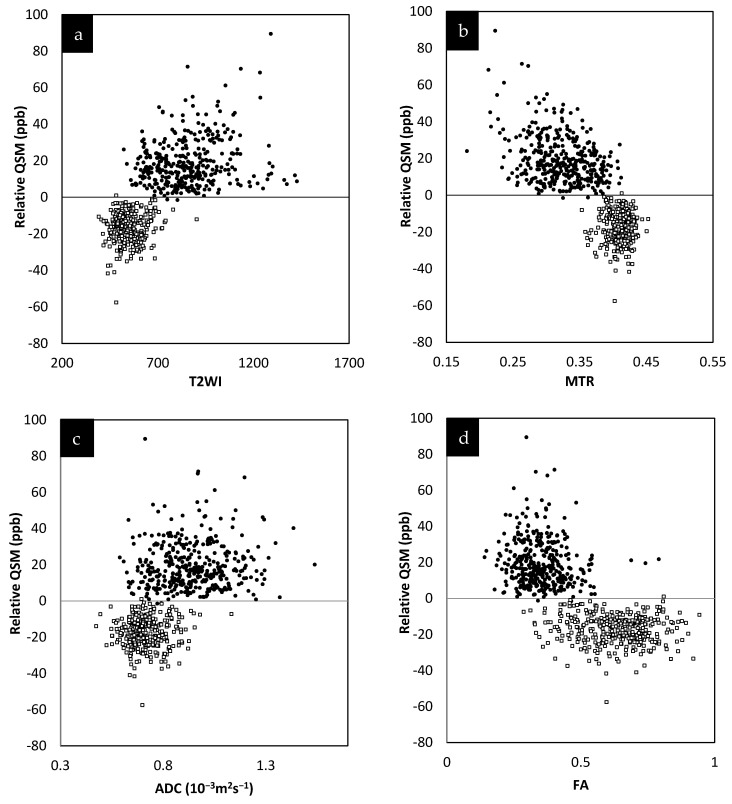
Relative susceptibility in parts per billion (ppb) for QSM versus MTR averaged across the two different time points (**a**). Relative susceptibility for QSM versus T2WI averaged across the two different time points (**b**). Relative susceptibility for QSM versus ADC averaged across the two different time points (**c**). Relative susceptibility for QSM versus FA averaged across the two different time points (**d**). In all these plots, the results for the MS lesion (solid black circles) values are well separated from the normal tissue values (square sign).

**Figure 5 diagnostics-12-00077-f005:**
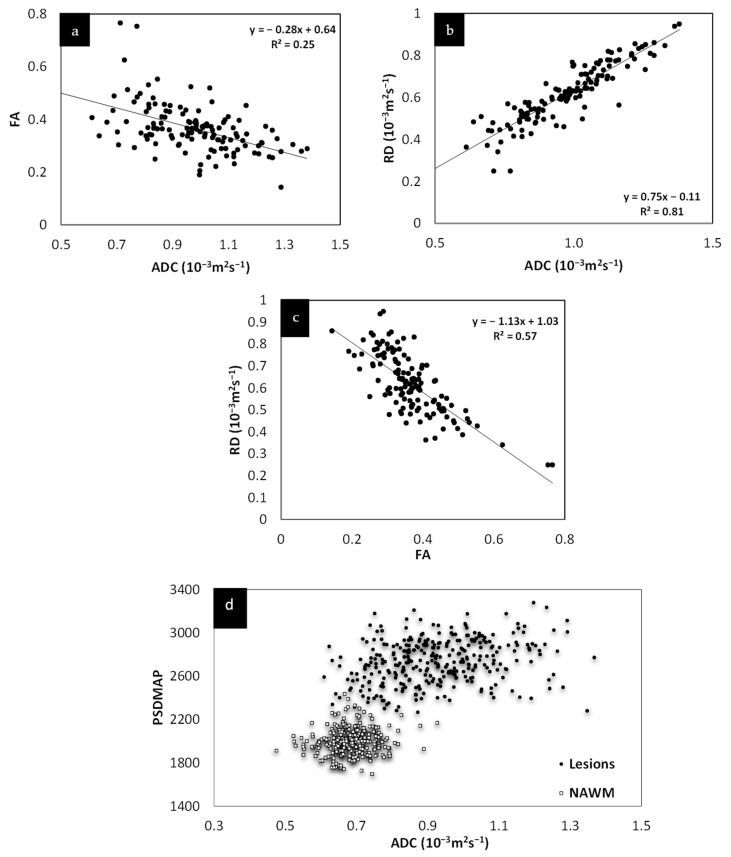
Relationship between FA, RD and ADC. Both FA and RD correlate with ADC (**a**,**b**). Although both FA and RD correlate with each other (**c**) and with ADC, RD appears to show the best correlation with ADC. The fact that these various measures correlate with each other and, as shown in (**d**), that ADC correlates with spin density, all this suggests that these values are driven by the same mechanism, increased water content.

**Figure 6 diagnostics-12-00077-f006:**
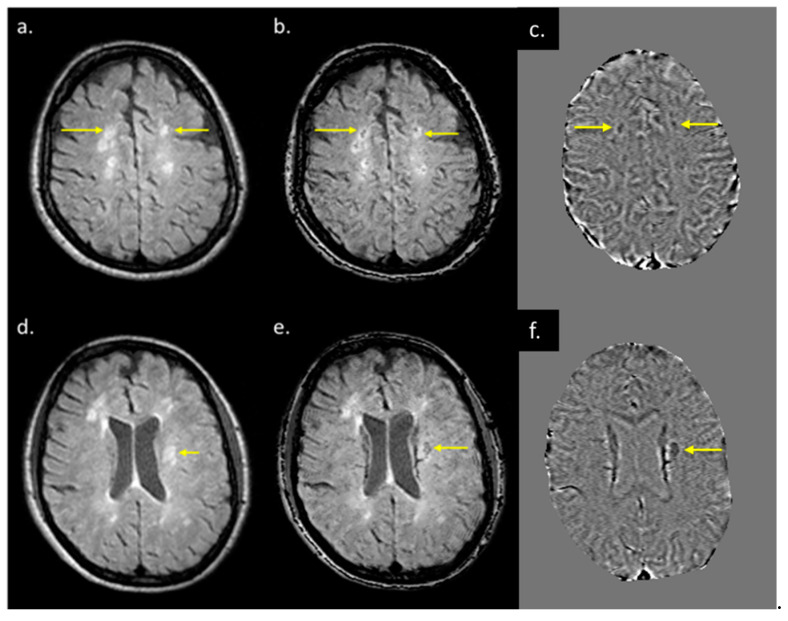
T2 FLAIR images (**a**,**d**), fusion image of susceptibility weighted imaging (SWI) phase and T2 FLAIR (**b**,**e**), and SWI phase (**c**,**f**) from two patients both with QSM and T2 FLAIR visible lesions. When T2 FLAIR images (**a**,**d**) have an SWI phase mask superimposed on them (**c**,**f**), possible regions of demyelination (shown by yellow arrows in (**b**,**e**) can be differentiated from those showing inflammation alone (**b**,**e**).

**Figure 7 diagnostics-12-00077-f007:**
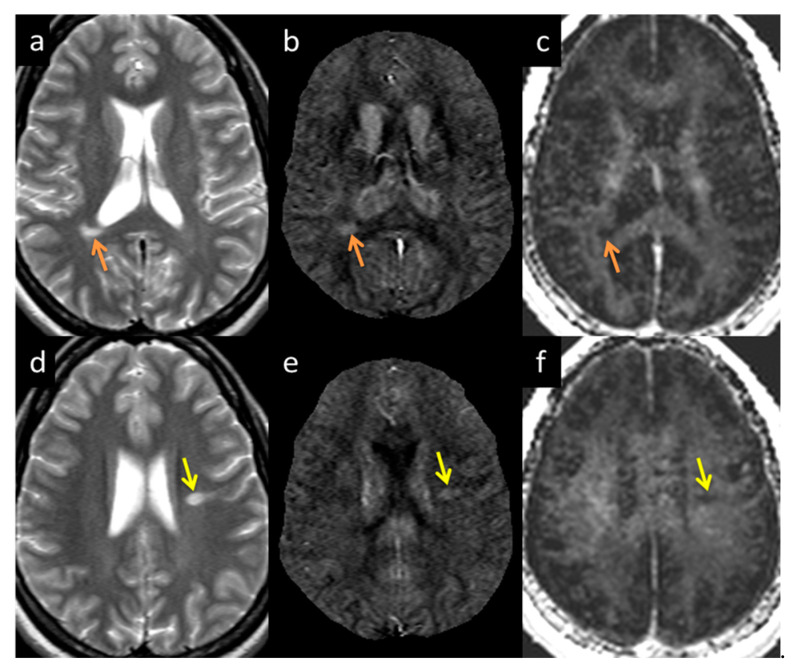
Lesion appearance example in two different subjects: Subject 1 (**a**–**c** denoted by the orange arrow) and Subject 2 (**d**–**f** denoted by the yellow arrow) in T2WI (**a**,**d**), QSM (**b**,**e**), and MWF (**c**,**f**). The lesions appearing in MWF (**c**,**f**) also appear bright in QSM (**b**,**e**) with high susceptibility directly correlating to demyelination.

**Figure 8 diagnostics-12-00077-f008:**
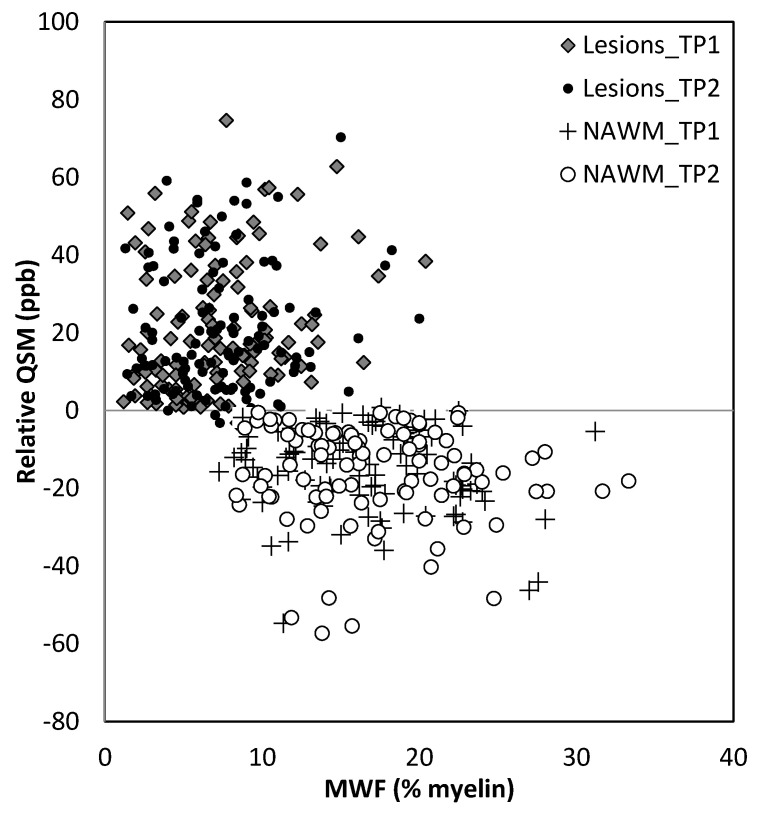
Relative QSM versus myelin water fraction (MWF). Lesions appear to have different susceptibilities and MWF for lesions compared to normal appearing white matter (NAWM). TP1: time point 1; TP2: time point 2.

**Figure 9 diagnostics-12-00077-f009:**
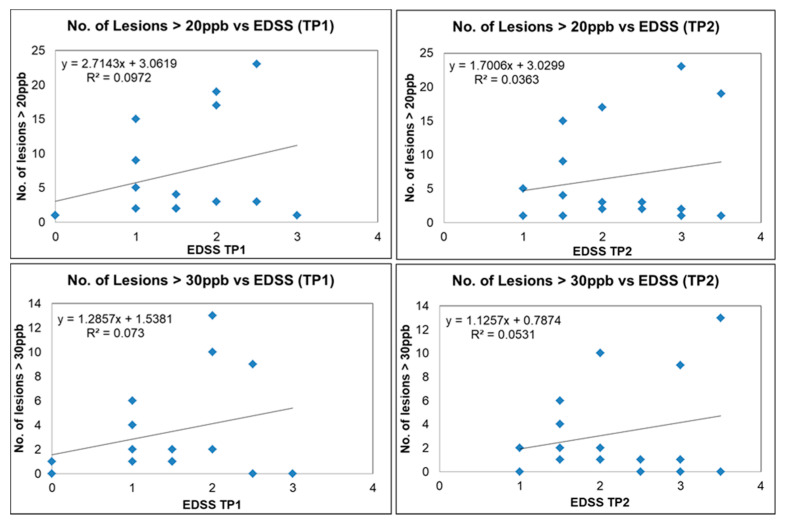
Number of lesions versus EDSS for QSM-positive lesions for different MS subjects (blue squares) with susceptibility of more than 20 ppb for time point 1 (upper left image) and time point 2 (upper right image). Number of lesions versus EDSS for QSM-positive lesions with susceptibility more than 30 ppb for time point 1 (lower left image) and time point 2 (lower right image).

## Data Availability

These data are not publicly available.

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
