# Peer review of "A Comparison of Magnetic Resonance Imaging Methods to Assess Multiple Sclerosis Lesions: Implications for Patient Characterization and Clinical Trial Design"

_diagnostics, 2021, doi:10.3390/diagnostics12010077_

Round 1

Reviewer 1 Report

I have reviewed the following manuscript “A comparison of magnetic resonance imaging methods to assess multiple sclerosis lesions: Implications for patient characterization and clinical trials design” I have following comments:

In this study, authors tried to evaluate and compare different types of MRI imaging (such as T1W1,  T2W1 weighted imaging, T2 fluid attenuated inversion recovery (FLAIR), magnetization transfer, myelin water fraction, diffusion tensor imaging (DTI), phase sensitive inversion recovery and susceptibility weighted imaging (SWI)) or the clinical diagnosis of MS and evaluating drug response.

This study has combined and correlated different MRI-related imaging modalities in detecting MS-related lesions. Investigating the best possible imaging approach to detect and diagnose MS-related lesions can take a huge burden off of the patients, who have to go thought various imaging evaluation.  

This study claims that imaging protocol combining T2 FLAIR, T1WI and STAGE (with SWI and QSM) can be used to rapidly image MS patients to both find lesions and to study the demyelinating and inflammatory characteristics of the lesions.

The data, shown in the manuscript does support the conclusion and it may be helpful in future in better diagnosing MS lesions in the patients without having them undergoing several detection methods.

English language and grammer should be proofread. I recommend this for the consideration of publication.

Author Response

We thank the reviewer for their thoughtful review of our work and recommendation. We have thoroughly reread the paper to check for any spelling or grammatical errors.

Reviewer 2 Report

This article describes the results of a study conducted on multiple sclerosis imaging which have compared different MRI measures for lesion visualization and quantification. The article in general is well written and doesn’t need English editing. The topic is overall interesting and could provide a clinically-relevant tool for managing multiple sclerosis imaging, giving useful suggestion also for a practical imaging protocol since the use of sequences with redundant meaning could be avoid. However, some concerns need to be acknowledged:

- The introduction should be improved since it is too short. At the end of the introduction the authors state: “ ..patients were imaged twice to evaluate stability and reproducibility of these measures and to correlate the various qualitative and quantitative measures with patient outcomes”. I haven’t found any correlation between measures and patient outcome in the results. Please clarify and specify better the purposes of the study and what do you mean for patient outcome.

- The imaging section in the methods is clearly and deeply described but the statistic section in my opinion should be improved.

At the end of the paper a separate section to describe limits would be better, and should be also improved since there are more limitations that need to be discussed. Take care to discuss drawbacks of the study.

- The patient cohort is small. What else might bias these results?

-What would enable better definition of water content?
